# Semi-Supervised Neural Network Model For Quadratic Multiparametric Programming

## Abstract

Neural Networks (NN) with ReLU activation functions have been used to model multiparametric quadratic optimization problems (mp-QP) in diverse engineering applications. Researchers have suggested leveraging the piecewise affine property of deep NN models to solve mp-QP with linear constraints, which also exhibit piecewise affine behaviour. However, traditional deep NN applications to mp-QP fall short of providing optimal and feasible predictions, even when trained on large datasets. This study proposes a semi-supervised NN (SSNN) architecture that directly represents the mathematical structure of the global solution function. In contrast to generic NN training approaches, the proposed SSNN method derives a large proportion of model weights directly from the physical characteristics of the system, producing more accurate solutions despite significantly smaller training data sets. Since many energy management problems are formulated as QP, the proposed approach has been applied in energy systems to demonstrate proof of concept. Model performance in terms of solution accuracy and speed of predictions was compared against a commercial solver and a generic DNN model based on classical training. Results show KKT sufficient conditions for SSNN consistently outperform generic NN architectures with classical training using far less data, including when tested on extreme, out-of-training distribution test data. Given its speed advantages over traditional solvers, the SSNN model can quickly produce optimal and feasible solutions within a second for millions of input parameters sampled from a distribution of stochastic demands and renewable generator dispatches, which can be used for simulations and long term planning.

## 1 Introduction

There has been increasing interest in using Neural Networks (NN) to predict the solutions to complex nonlinear optimization problems in energy management, chemistry, control theory, and other domains. Effectively, this goal amounts to estimating a global solution function [1] that predicts optimal solutions based on a given set of input parameters, such as the right hand side (RHS) vector and cost coefficients of an optimization model. Most applications treat the NN model as a black-box and employ standard training methods using datasets of input-output pairs, representing particular values of system parameters and corresponding optimal solutions. Most also ignore the mathematical structure of the underlying function and utilize a generic NN architecture to approximate solutions. Such modelling approaches necessitate large, computationally expensive training datasets to achieve satisfactory performance, yet cannot guarantee the feasibility or optimality of results.

The multiparametric programming literature similarly seeks to represent the global solution function to optimization problems by characterizing the optimal solutions as a function of feasible problem parameters, and identifying the regions of parameter space where these functions are valid (Pistikopoulos et al., 2020). Recent work has attempted to integrate multiparametric programming with DNNs, based on recognition of their similar underlying mathematical characteristics (Karg & Lucia, 2020; Katz et al., 2020). Specifically, for multiparametric Linear Programming (mp-LP) and Quadratic Programming (mp-QP) with linear constraints, the optimal primal and dual solutions are piecewise linear (PWL) functions of the parameters (Pistikopoulos et al., 2020). Similarly, a NN

---

[1]The function of optimal primal solutions with respect to parameters is sometimes called optimizer in the literature. In our study, solution function is used to refer to the function of both primal and dual solutions.

with ReLU activation functions is a general form of PWL function with trainable weights and biases (Montufar et al., 2014). In theory, therefore, for any mp-LP or mp-QP, one should be able to find an optimal set of weights and biases such that a NN can exactly represent the corresponding solution function (Karg & Lucia, 2020).

In practice, however, DNN applications of multiparametric programming face significant training challenges to guarantee global solution optimality and feasibility. First, obtaining the optimal NN weights and biases is difficult due to nonconvexity of the training loss function, whereby multiple sets of weights and biases can result in different local minima for a given dataset. Second, NN prediction accuracy is highly dependent on the training set, which makes it hard to generalize predictions outside the training distribution. Overcoming this problem to find the global solution function requires sampling very large datasets that cover all feasible regions of the parameters, which is impractical for problems with high dimensions. Specifically, for an $n$ dimensional domain, constructing a dataset that includes only boundary points for each feasible region requires sampling $2^n$ points, which would still be insufficient because additional interior points are needed within each region for good estimation.

To address these concerns, this paper proposes a semi-supervised NN (SSNN) architecture and training procedure for mp-QP problems that directly reflects the mathematical structure of the global solution function. The proposed approach dramatically reduces training dataset demands, while satisfying sufficient optimality and feasibility requirements. In general, while the solution function to mp-QP for uncertain parameters is piecewise linear, the slope changes in the solution function can be decomposed into two components: piecewise linear changes in the value of dual variables associated with inequality constraints, and the remaining terms that form a linear function. The proposed SSNN implements these components separately in a single NN architecture, whose calculations can be easily parallelized using GPU to predict solutions for a large set of input parameters. The proposed SSNN approach was used to model DC optimal power flow (DC-OPF) problems in the domain of electricity power management for proof-of-concept support.

Unlike black-box NN models, the proposed SSNN for QP is largely constructed unsupervised by deriving NN weights directly from the problem. The coefficients are calculated prior to the training by expanding the Lagrangian function with each possible combination of binding constraints, calculating partial derivatives and inverting the resulting Jacobian matrices of the derivatives of the Lagrangian function. Theoretically, the prior computation of coefficients can be infeasible for large problems, as the number of constraint combinations increases exponentially. Nevertheless, in practical applications, many constraint combinations are not possible due to characteristics of the problem, such as the topology of the electricity grid in energy management. To scale down the set of all possible binding constraint combinations, we use a commercial solver to solve the problem for a set of input system parameters sampled equidistantly from minimum to maximum feasible input demand for a number of arbitrarily chosen parameters. Note that this procedure is used only to filter out unused cases and is run only about 500 times for even the largest empirical test systems investigated. Once the set of possible binding constraints is discovered, this information is used to construct datasets with minimal computational costs without using solvers in a loop.

Our study makes the following contributions:

- A semi-supervised NN model is proposed to provide optimal solutions to QP with inequality constraints. The model is based on an explainable architecture that aligns with the piecewise linearity of the global solution function, and is supported by a training approach that decomposes the learning process to sub-problems to overcome the training challenges.
- The proposed model generalizes outside the training distribution well, thanks to a procedure that samples model weights analytically from the underlying optimization problem.
- The model is applied to DC-OPF with upper and lower bound generator limits, and provides precise predictions that satisfy KKT optimality and feasibility conditions. The model can rapidly create a distribution of optimal and feasible solutions to a large dataset sampled from an empirical distribution of uncertain demands and renewable generator dispatches.

The paper is organized as follows: Section 2 discusses related studies in the literature. Section 3 details our methodology, discusses how QP solutions form a PWL function that can be decomposed to linear and piecewise parts, and outlines our semi-supervised NN architecture and training procedure. Section 4 presents numerical results, and section 5 discusses conclusions and future work.

## 2 RELATED WORK

Using NNs for surrogate modeling of optimization problems is an established field with various applications in engineering. The approaches leverage DNNs as universal function approximators to estimate the nonlinear relationship between the problem parameters and the optimal solutions, which has found success in power management (Nellikkath & Chatzivasileiadis, 2022; Fioretto et al., 2020; Lotfi & Pirnia, 2022), control Karg & Lucia (2020) and chemistry (see Katz et al. (2020) for a review). Despite their advantages as nonlinear solvers, guaranteeing feasability and optimality is challenging, and finding the best model parameters from a large parameter space is data-greedy with significant computational training burdens.

To improve feasibility, Zamzam & Baker (2020) and Chen et al. (2023) proposed two-stage models that first predict optimal solutions with a NN and then post-process the solutions to enforce feasibility. Lotfi & Pirnia (2022) proposed a NN architecture that enforces optimal dispatch generator limits using a sigmoid activation function in the output layer. Fioretto et al. (2020) processed optimal solutions through sequentially connected sub-networks reflecting problem characteristics. Physics Informed Neural Networks (PINNs) use DNNs that are also trained to satisfy KKT optimality conditions, resulting in reduced dataset requirements, more accurate predictions with less violations, and more generalizability power (Nellikkath & Chatzivasileiadis, 2022). Instead of predicting optimal solutions directly, others proposed training different types of classifiers to predict active constraints, thereby reducing the problem dimensions, providing more accurate predictions and quicker solutions (e.g. Misra et al. (2021); Deka & Misra (2019); Ng et al. (2018); Chen et al. (2022)).

Examining mathematical properties of mp-LP and mp-QP problems under a multiparametric setting led researchers to recognize their similarity to the PWL characteristics of DNNs. Katz et al. (2020) attempted to integrate PWL behaviour of the problem solution analytically. Karg & Lucia (2020) examined the functional similarities between DNNs and model predictive control (MPC) laws, and trained a model to estimate MPC solutions for a time variant system. Similarly, Huo et al. (2022) proposed an approach to integrate MPC laws of a MILP problem for a microgrid system. However, none of these works compare optimality and feasibility between their models and traditional solver benchmarks or DNN based approaches. This paper proposes a custom NN architecture that closely aligns with the PWL structure of the global solution function to improve accuracy and generalizability of mp-QP predictions.

## 3 A GENERAL FRAMEWORK TO SOLVE QP USING NN

In general form, mp-QP with linear constraints can be written as,

$$\textbf{Minimize } \mathbf{F}(\boldsymbol{\theta}) = \mathbf{x}^T\mathbf{Q}\mathbf{x} + (\mathbf{C} + \mathbf{F}_c\boldsymbol{\theta})^T\mathbf{x} + \mathbf{C}_0, \tag{1a}$$

$$\textbf{s.t. } \mathbf{A}_e\mathbf{x} = \mathbf{b}_e + \mathbf{F}_e\boldsymbol{\theta}, \quad [\lambda] \tag{1b}$$

$$\mathbf{A}_{\mathcal{C}}\mathbf{x} \leq \mathbf{b}_{\mathcal{C}} + \mathbf{F}_{\mathcal{C}}\boldsymbol{\theta}, \quad [\mu] \tag{1c}$$

where $\mathbf{x}, \mathbf{C}, \mathbf{C}_0 \in \mathbb{R}^n, \mathbf{b}_e \in \mathbb{R}^{m_1}, \mathbf{b}_{\mathcal{C}} \in \mathbb{R}^{m_2}, \mathbf{Q} \in \mathbb{R}^{n \times n}$ is a semi-positive definite matrix, and $\mathbf{A}_e \in \mathbb{R}^{m_1 \times n}, \mathbf{A}_{\mathcal{C}} \in \mathbb{R}^{m_2 \times n}$ and $\boldsymbol{\theta} \in \Theta_f$ where $\Theta_f$ is the set of all feasible parameters. The problem is parametrized by $\boldsymbol{\theta} \in \mathbb{R}^k$ for $k = n + m_1 + m_2$ and the coefficients $\mathbf{F}_c, \mathbf{F}_e, \mathbf{F}_{\mathcal{C}}$ are controlling sensitivity of $\boldsymbol{\theta}$ on changes in linear cost term and right hand side vectors. The Lagrangian function of the above problem can be written as,

$$L(\mathbf{x}, \boldsymbol{\lambda}, \boldsymbol{\mu}) = \mathbf{x}^T\mathbf{Q}\mathbf{x} + (\mathbf{C} + \mathbf{F}_c\boldsymbol{\theta})^T\mathbf{x} + \mathbf{C}_0 + \boldsymbol{\lambda}^T(\mathbf{b}_e + \mathbf{F}_e\boldsymbol{\theta} - \mathbf{A}_e\mathbf{x}) + \boldsymbol{\mu}^T(\mathbf{b}_{\mathcal{C}} + \mathbf{F}_{\mathcal{C}}\boldsymbol{\theta} - \mathbf{A}_{\mathcal{C}}\mathbf{x}), \tag{2}$$

where $\boldsymbol{\lambda} \in \mathbb{R}^{m_1}, \boldsymbol{\mu} \in \mathbb{R}^{m_2}$ are the dual variables associated with equality and inequality constraints, respectively.

If the solution to the subproblem (1a)-(1b) satisfies (1c), the corresponding shadow prices must be zero at optimality, i.e. $\boldsymbol{\mu}^* = 0$. However, when the solution violates (1c), it can be attained by adding one or more violated constraints iteratively to reach the optimal solution. In the following sections, the general cases of optimization problems with equality constraints (1b) and inequality constraints (1c) are considered, and methods for estimating the dual variables are proposed. Finally, a NN based solution method to such optimization problems is suggested.

### 3.1 Solution to Equality Constrained mp-QP

For the case when constraint (1c) is not binding, $\boldsymbol{\mu}^* = 0$ and the Lagrangian function (2) can be written as follows.

$$L(\mathbf{x}, \boldsymbol{\lambda}, \boldsymbol{\mu}) = \mathbf{x}^T \mathbf{Q} \mathbf{x} + (\mathbf{C} + \mathbf{F}_c \boldsymbol{\theta})^T \mathbf{x} + \mathbf{C}_0 + \boldsymbol{\lambda}^T (\mathbf{b}_e + \mathbf{F}_e \boldsymbol{\theta} - \mathbf{A}_e \mathbf{x}). \tag{3}$$

Using the fact that the gradient of the Lagrangian function (3) with respect to $\mathbf{x}, \boldsymbol{\lambda}$ must be zero at optimality, equation (4) defines a function that produces optimal solutions to any given right hand side vector, $\mathbf{b}_e$ (see Appendix A.2 for derivation).

$$g(0; \boldsymbol{\theta}) = \mathbf{J}^{-1} \begin{bmatrix} -\mathbf{C} - \mathbf{F}_c \boldsymbol{\theta} \\ -\mathbf{b}_e - \mathbf{F}_e \boldsymbol{\theta} \end{bmatrix}, \quad \text{where} \quad \mathbf{J} = \begin{bmatrix} 2\mathbf{Q} & -\mathbf{A}_e^T \\ -\mathbf{A}_e & 0 \end{bmatrix}, \tag{4}$$

represent the coefficient matrix. Equation (4) is a function yielding the optimal solutions for (1a)-(1b) as $g(0; \boldsymbol{\theta}) = [\mathbf{x}^*, \boldsymbol{\lambda}^*]$, which will be referred to as *solution function* in the rest of the study. Here the 0 in the $g(\cdot)$ function is the value of the shadow price, $\boldsymbol{\mu}^* = 0$, that will be used to generalize to inequality constrained problems in the next section.

### 3.2 Adding Inequality Constraints

As discussed earlier, in the case when inequality constraints are binding, the active constraints can be added as equality constraints while the optimal dual variables of inequality constraints are positive, i.e. $\boldsymbol{\mu}^* > 0$. Then, the solution can be found by expanding the Lagrangian function and finding its critical points.

Assume that for any system parameter, $\boldsymbol{\theta}$, the set of indices of binding constraints $\mathcal{B} \subseteq \{1, \ldots, m_2\}$ is known. Then, the optimal solution to the problem can be found by solving

$$\frac{\partial L}{\partial \mathbf{x}} = 2\mathbf{Q}\mathbf{x} + \mathbf{C} + \mathbf{F}_c \boldsymbol{\theta} + \boldsymbol{\lambda}^T \mathbf{A}_e + \boldsymbol{\mu} = 0 \tag{5a}$$

$$\frac{\partial L}{\partial \boldsymbol{\lambda}} = \mathbf{b}_e + \mathbf{F}_e \boldsymbol{\theta} - \mathbf{A}_e \mathbf{x} = 0, \tag{5b}$$

$$\frac{\partial L}{\partial \boldsymbol{\mu}} = \mathbf{b}_{\mathcal{B}} + \mathbf{F}_{\mathcal{B}} \boldsymbol{\theta} - \mathbf{A}_{\mathcal{B}} \mathbf{x} = 0, \tag{5c}$$

where $\mathbf{A}_{\mathcal{B}} = \{\mathbf{A}_{\mathcal{C}}\}_{i \in \mathcal{B}}, \mathbf{b}_{\mathcal{B}} = \{\mathbf{b}_{\mathcal{C}}\}_{i \in \mathcal{B}}$ are matrices consisting of binding constraints, and optimal shadow prices $\mu_i^* > 0 \ \forall i \in \mathcal{B}$ and $\mu_i^* = 0 \ \forall i \notin \mathcal{B}$.

Now, if the function $\boldsymbol{\mu}(\boldsymbol{\theta})$ that generates $\boldsymbol{\mu}^*$ is known, the remaining optimal solutions can be attained by modifying (2) and writing the derivatives as

$$\frac{\partial L}{\partial \mathbf{x}} = 2\mathbf{Q}\mathbf{x} + \mathbf{C} + \mathbf{F}_c \boldsymbol{\theta} + \boldsymbol{\lambda}^T \mathbf{A}_e + \boldsymbol{\mu}(\boldsymbol{\theta}) = 0, \tag{6a}$$

$$\frac{\partial L}{\partial \boldsymbol{\lambda}} = \mathbf{b}_e + \mathbf{F}_e \boldsymbol{\theta} - \mathbf{A}_e \mathbf{x} = 0. \tag{6b}$$

The above mapping is the same as the equality constrained solution function in 4. Therefore, if $\boldsymbol{\mu}(\boldsymbol{\theta}) = \boldsymbol{\mu}^*$ is given, the optimal solution can be obtained using the solution function,

$$g(\boldsymbol{\mu}^*; \boldsymbol{\theta}) = [\mathbf{x}^*, \boldsymbol{\lambda}^*]^T. \tag{7}$$

As the solution function is linear, the only nonlinearity can stem from $\boldsymbol{\mu}^*$. The next section discusses the piecewise linear nature of $\boldsymbol{\mu}^*$ and how it can be estimated exactly with a NN.

### 3.3 Predicting Active Constraints and Shadow Prices

As shown in (7), estimating the solution of the optimization problem (1a)-(1b) requires knowing the value of $\boldsymbol{\mu}^*$. Therefore, in this section, a methodology to produce exact predictions of $\boldsymbol{\mu}^*$ and hence the active constraints is presented. While $g(\cdot)$ in eq. (7) is a linear function, the addition of $\mu^*$ changes the slope of the solutions with respect to $\mathbf{b}_1$ and converts the function to PWL due to its piecewise linear nature.

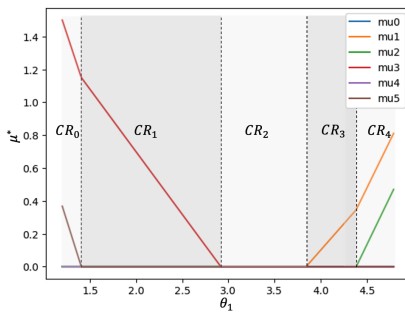

Figure 1: Sensitivity of $\boldsymbol{\mu}$ against $\theta_1$

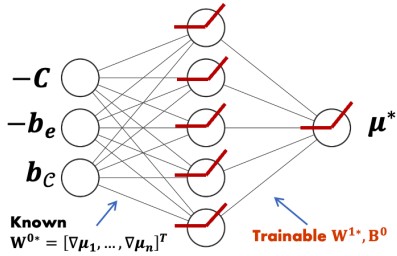

Figure 2: NN Model to Predict $\boldsymbol{\mu}^*$

**Theorem 1.** *For the mp-QP problem 1, $\Theta_f \subseteq \Theta$ is a convex set, the primal solution function $\boldsymbol{x}(\boldsymbol{\theta})$ : $\Theta_f \to \mathbb{R}^n$ is continuous and piecewise affine. Also the optimal objective function $\boldsymbol{F}(\boldsymbol{\theta}) : \Theta_f \to \mathbb{R}$ is continuous and piecewise quadratic.*

*Proof:* The reader can find the proof of the theorem in Pistikopoulos et al. (2020).

**Lemma 1.** *Consider the mp-QP problem 1. The shadow prices of inequality constraints, $\boldsymbol{\mu}^*$ : $\Theta_f \to \mathbb{R}^{m_2}$ are piecewise affine.*

*Proof:* The mapping $g(\cdot)$ in eq. 7 is an affine mapping from $\boldsymbol{\mu}^* = \mu(\boldsymbol{\theta})$, and as shown in Theorem 1, the output of $\mathbf{x}(\boldsymbol{\theta})$ is a piecewise affine function of $\boldsymbol{\theta}$. As $g(\cdot)$ can be defined as a composite function, i.e. $(g \circ \mu)(\boldsymbol{\theta})$, the only input of the function has to be piecewise affine.

To illustrate, consider an optimization problem in the form defined in (1), with 5 decision variables, $x_i$, three equality constraints, and three upper and lower bound constraints; i.e. $x_i^- \leq x_i \leq x_i^+$ for $i \in [0, 1, 2]$. As will be shown later, this is our smallest 6-bus DC-OPF application case defined in eq. 9. Here, the problem is solved using a Gurobi solver (v10.0.3) for $\theta_1 \in [0, 0.1, 0.2, \ldots, 5]$ controlling RHS of equality constraints $\mathbf{b}_e + \mathbf{F}_e \boldsymbol{\theta} = [\theta_1, .001, .001]$. In Figure 1, the optimal dual variables are plotted against different values of $\theta_1$. $\mu_i$ for $i \in [0, 1, 2]$ and $\mu_i$ for $i \in [3, 4, 5]$ are shadow prices of the upper and lower limits, respectively, specifically the limits of power generators in the example. The slope changes in $\boldsymbol{\mu}^*$ exhibit a piecewise affine pattern, with different slopes in different critical regions (**CR**), corresponding to different combinations of active constraints. In $\mathbf{CR}_0$ two lower limit constraints are active with corresponding $\mu_i > 0$. As $\theta_1$ increase, generator lower limits stop binding and upper limits start to bind.

While $\boldsymbol{\mu}^*$ as a function of $\boldsymbol{\theta}$ is piecewise linear and can be modelled with a generic NN, training a model on a dataset consisting of $(\boldsymbol{\theta}, \boldsymbol{\mu}^*)$ pairs does not guarantee optimality of the predictions for test data sampled outside this distribution. To overcome this difficulty, our method directly injects some of the information that is general to all $\boldsymbol{\theta} \in \Theta_f$, into the first layer of the model. Specifically, all the slopes that the $\mu$-NN will have for each case of binding constraints are calculated and fixed as the first layer weights prior to the training, then the rest of the model parameters, biases and second layer weights, are estimated via training. Figure 2 illustrates the architecture of this model where $\mathbf{W}^0$ indicates the injected weights for the first layer, obtained from $\nabla \boldsymbol{\mu}_{\mathcal{B}}^*$ as follows:

*Calculation of $\boldsymbol{\mu}_{\mathcal{B}i}^*$:* To obtain the slopes of $\boldsymbol{\mu}^*$ with respect to the changes in $\boldsymbol{\theta}$ for binding constraints, $\nabla \boldsymbol{\mu}_{\mathcal{B}_i}^*$ we expand the $\mathbf{J}$ matrix defined in eq. 4 with the rows of the coefficient matrix of inequality constraints corresponding to the binding constraints, $\mathbf{A}_{\mathcal{B}_i}$, and then inverting it.

$$\begin{bmatrix} \mathbf{x}^* \\ \boldsymbol{\lambda}^* \\ \boldsymbol{\mu}_{\mathcal{B}_i}^* \end{bmatrix} = \mathbf{J}_{\mathcal{B}_i}^{-1} \begin{bmatrix} -\mathbf{C} \\ -\mathbf{b}_e \\ \mathbf{b}_{\mathcal{B}_i} \end{bmatrix} = \begin{bmatrix} \nabla \mathbf{x}^* \\ \nabla \boldsymbol{\lambda}^* \\ \nabla \boldsymbol{\mu}_{\mathcal{B}_i}^* \end{bmatrix} \begin{bmatrix} -\mathbf{C} \\ -\mathbf{b}_e \\ \mathbf{b}_{\mathcal{B}_i}. \end{bmatrix}, \text{ where } \mathbf{J}_{\mathcal{B}} = \begin{bmatrix} 2\mathbf{Q} & -\mathbf{A}_e^T & \mathbf{A}_{\mathcal{B}_i}^T \\ -\mathbf{A}_e & 0 & 0 \\ \mathbf{A}_{\mathcal{B}_i} & 0 & 0 \end{bmatrix}. \quad (8)$$

Notice that the last row of the above inverse Jacobian matrix, $\mathbf{J}_{\mathcal{B}_i}^{-1}$, is the gradient vector $\nabla \boldsymbol{\mu}_{\mathcal{B}_i}^*$ that contains the coefficients of the linear facet of the target PWL function when a certain set of constraints, $\mathcal{B}_i$, are binding. In other words, the vector is the set of coefficients that maps $[\mathbf{C}, \mathbf{b}_e, \mathbf{b}_{\mathcal{B}_i}]$ to $\boldsymbol{\mu}_{\mathcal{B}_i}^*$.

After the first layer of the NN is fixed with all potential derivatives, the training is carried out to find when each slope is activated. However, the number of potential slopes increases exponentially with the addition of inequality constraints, which adds more burden on the pre-training step as matrix inversion is required to calculate the slopes and also increases the number of parameters to estimate in the training step. The next section outlines a method to calculate $\nabla\boldsymbol{\mu}_{\mathcal{B}_i}$ without matrix inversion and a method to filter active constraints to enhance the execution time of the solution methodology.

### 3.4 FILTERING ACTIVE CONSTRAINTS AND DATA GENERATION

As explained in the Introduction, in many practical mp-QP applications, characteristics of the problem imply that most constraint combinations can never occur, so some method of filtering out the unused combinations is needed. For example, in the previous 6-bus DC-OPF example with 6 inequality constraints, the set of all constraint combinations is $\{\{0\}, \{1\}, \{2\}, \{0, 1\}, \dots, \{0, 1, 2, 3, 4, 5\}\}$ with $2^6 = 64$ elements, yet 59 of these combinations never occur. In the present study, to identify the active combinations we use a commercial solver to solve the problem for a wide range of input parameters. Specifically, the RHS vector is set to $\mathbf{b}_e = \mathbf{0}$ and an arbitrary dimension is incrementally increased from zero to the highest feasible point. Then, after solving for each input demand, the set of potential binding constraints is identified by checking whether the associated $\mu_i$ is greater than zero for any of the inputs.

Using a solver to find potential binding constraints also gives insight into the boundaries of critical regions, $[\mathbf{CR}_i^-, \mathbf{CR}_i^+]$ corresponding to $\mathcal{B}_i$ that is active at solution. This information is used to populate data without using solvers, by sampling random inputs from $[\mathbf{CR}_i^-, \mathbf{CR}_i^+]$ for all $i$, obtaining the matrices $\mathbf{J}_{\mathcal{B}_i}^{-1}$ and $\mathbf{b}_{\mathcal{B}_i}$ following eq. 8, and calculating the corresponding optimal solutions.

The training procedure can be summarized as follows (see Appendix A.5 for further details):

1. Generate RHS and solve with a solver: (i) Initiate the RHS with an initial point, (ii) Choose a $\theta_i$ and increase its value from 0 to max.

2. Solve the problem and find $\mathbf{CR}_i$ and $\nabla\boldsymbol{\mu}_{\mathcal{B}_i}^*$: (i) Solve the problem for each generated data using Gurobi, (ii) Find the sets of active constraints, i.e. $\mathcal{B}_i = \{\, j \mid \mu_j > 0 \,\}$, (iii) Expand the coefficient matrix and calculate $\mathbf{J}_{\mathcal{B}_i}^{-1}$ for all binding constraints. Obtain $\nabla\boldsymbol{\mu}_{\mathcal{B}_i}^*$.

3. Populate data: Randomly generate $d_d$ number of RHS vectors using $Uniform(\mathbf{CR}_i^-, \mathbf{CR}_i^+)$ and find optimal solutions following 8. Check if KKT is satisfied for each data and regenerate data if needed.

4. Train $\mu$-NN: (i) Initiate $\mu$-predictor NN with random $\mathbf{W}^1, \mathbf{B}^0$, fix $\mathbf{W}^0 = [\boldsymbol{\mu}_{\mathcal{B}_1}^*, \dots, \boldsymbol{\mu}_{\mathcal{B}_n}^*]^T$ and set with random weights, (ii) Train the $\mu$-predictor NN with 1000 datapoints populated using the procedure described above, (iii) Define solver NN by setting first part to $\mu$-NN and second part to $\mathbf{J}^{-1}$.

## 4 NUMERICAL RESULTS

Optimal Power Flow (OPF) is a central problem in power system management (Carpentier, 1979). The problem is to find the optimal dispatch per generator to minimize an objective function (e.g., cost of operation) while satisfying energy demands and grid system constraints at a given point in time. The original AC-OPF problem is nonlinear and nonconvex, so challenges in finding optimal and feasible solutions led researchers to use the simplified DC problem that has linearized power flow constraints under certain assumptions. The DC-OPF problem can be defined as:

$$\underset{\mathbf{P}_g, \boldsymbol{\theta}}{\textbf{Minimize: }} \mathbf{P}_g^T \mathbf{Q} \mathbf{P}_g + \mathbf{C}^T \mathbf{P}_g + \mathbf{C}_0, \tag{9a}$$

$$\textbf{s.t. } \mathbf{P}_d + \mathbf{F}_e \boldsymbol{\theta} - \mathbf{B}\boldsymbol{\delta} - \mathbf{P}_g = 0, \quad [\boldsymbol{\lambda}] \tag{9b}$$

$$\mathbf{P}_g - \mathbf{P}_g^+ \leq 0, \quad [\boldsymbol{\mu}^+] \tag{9c}$$

$$\mathbf{P}_g^- - \mathbf{P}_g \leq 0, \quad [\boldsymbol{\mu}^-] \tag{9d}$$

where $\mathbf{C}, \mathbf{C}_0 \in \mathbb{R}^{n_g}$ are cost coefficient vectors, $\mathbf{Q} \in \mathbb{R}^{n_g \times n_g}$ is the quadratic cost coefficient matrix, $\mathbf{P}_g^-, \mathbf{P}_g^+ \in \mathbb{R}^{n_g}$ are lower and upper generator limits, and $n_b, n_g$ are number of bus and

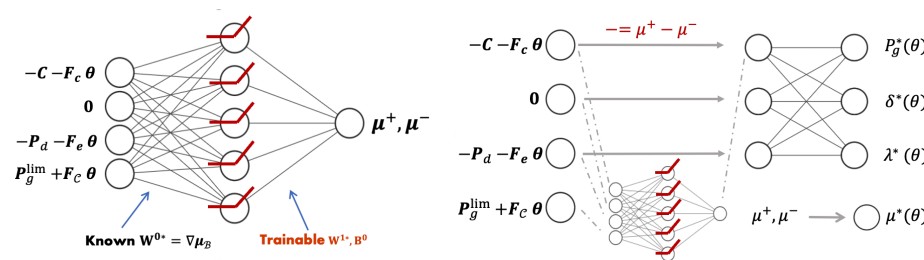

Figure 3: Submodel predicting $\boldsymbol{\mu}^*$        Figure 4: SSNN Architecture

generator buses, respectively. The primal variables are the production output of dispatchable generators, $\mathbf{P}_g, \in \mathbb{R}^{n_g}$ and voltage magnitudes of each bus, $\boldsymbol{\delta} \in \mathbb{R}^{n_b}$. Following the steps in Section 3, our model is derived using the derivatives of the Lagrangian function in eq. 5 for the DC-OPF problem. The reader can refer to Appendix A.3 for the derivation of the system of equations. After the derivations, the right hand side vectors translate into the demand vector, $\mathbf{b}_e = \mathbf{P}^{\text{lim}} = \mathbf{P}_d$ for equality constraints and the generator limits, $\mathbf{b}_{\mathcal{C}}^T = [\mathbf{P}_g^+, \mathbf{P}_g^-]$ for the inequality constraints, and the primal variables are $\mathbf{x}^T = [\mathbf{P}_g, \boldsymbol{\delta}]$.

We use a semi-supervised NN to predict the optimal solution through two sequential subnetworks, a shallow ReLU network defined in the form of eq. 12 of Appendix A.1, $\mu-$NN that calculates $\boldsymbol{\mu}^*$, and a separate NN that models $g(\boldsymbol{\mu}^*; \boldsymbol{\theta})$ in (4). Figure 3 illustrates the subnetwork that predicts the optimal dual variables of the inequality constraints, $\boldsymbol{\mu}^*$. As discussed earlier, the weights of the first layer of this subnetwork model are fixed with $\nabla \boldsymbol{\mu}_{\mathcal{B}_i}^*$ vectors for all potential binding constraints $i \in \mathcal{C}$. Originally, each of these slope coefficient vectors have different dimensions, depending on which constraints are binding, but they are modified to have the same size by adding columns or rows containing zeros. The layer weights are obtained simply by stacking all the vectors vertically. The rest of the parameters, weights of the second layer, and biases, are determined via training. Finally, the second subnetwork is a mapping $g(\boldsymbol{\mu}^*; \boldsymbol{\theta}) = [\mathbf{P}_g^*, \boldsymbol{\delta}^*, \boldsymbol{\lambda}^*]$, which can be defined as a linear model (one layered NN) without training, by fixing its weights to $\mathbf{W} = \mathbf{J}^{-1}$ defined in eq 4. Our SSNN model combines the two subnetworks in one NN flow as shown in Figure 4. The $\mu-$NN takes $\mathbf{C}, \mathbf{P}_d, \mathbf{P}^+, \mathbf{P}^-$ as inputs and produces $\boldsymbol{\mu}^*$. The second subnetwork takes the adjusted inputs to produce the optimal solutions.

### 4.1 Test Results

Our method was evaluated using IEEE-6, IEEE-30, IEEE-57 bus test systems. We used our SSNN model to predict optimal solutions to the DC-OPF problem, and compared them to solutions predicted by a NN model trained using classical methods (i.e., with paired input-output data representing particular values of system parameters and corresponding optimal solutions), and also to solutions generated by the Gurobi solver (v10.0.3). The SSNN models were trained using only 1000 data points, and validated and tested on datasets of 1000 and 4000 observations, respectively. To generate training data, demand of all buses was set to 0.01, then one bus was chosen and its load incrementally increased from 0 to the maximum allowed by the system, and the procedure was repeated for one other bus. For the classically trained NN model, we generated training, validation and test sets of 5000, 1000 and 4000 observations, respectively, by setting the demand to the default load values in the Pandapower package and multiplying with a constant sampled from Uniform$(0.6, 1.4)$. Two types of test datasets were generated using both approaches, reflecting realistic and extreme ranges of demand. Realistic demand data was generated by multiplying a base demand by a constant sampled from Uniform$(0.6, 1.4)$. Extreme demand data was generated by fixing all load bus demands to 0.01, replacing one bus with a number sampled from $Uniform(0, \max)$ and repeating this step for all buses. All the experiments and training were done on a Mac Mini M2 (2023) on Python 3.9.18, using PyTorch 2.0.1.

The SSNN, classically-trained DNN, and Gurobi solver solutions were compared based on violation of the KKT conditions, namely (i) Stationarity eq. (KKT 1 $\mathbf{P}_g, \boldsymbol{\delta}$), (ii) Primal feasibility (KKT 2), (iii) Dual feasibility (KKT 3) and (iv) Complementary slackness (KKT 4), which are given below.

| KKT 1 ($\mathbf{P}_g$) | KKT 1 ($\boldsymbol{\delta}$) | KKT 2 | KKT 3 | KKT 4 |
|---|---|---|---|---|
| $\left(\frac{\partial L}{\partial \mathbf{P}_g}\right)^2$ | $\left(\frac{\partial L}{\partial \boldsymbol{\delta}}\right)^2$ | $\left(\frac{\partial L}{\partial \boldsymbol{\lambda}}\right)^2$ | $\left[\max\left(0, \frac{\partial L}{\partial \boldsymbol{\mu}}\right)\right]^2$ | $\left(\boldsymbol{\mu}^* \frac{\partial L}{\partial \boldsymbol{\mu}}\right)^2$ |

Table 1: KKT Conditions and corresponding formulae

|  |  | KKT 1 ($\mathbf{P}^g$) | KKT 1 ($\boldsymbol{\delta}$) | KKT 2 | KKT 3 | KKT 4 |
|---|---|---|---|---|---|---|
| SSNN | 6 bus | 4.79E-13 | 1.46E-10 | 1.44E-13 | 1.49E-14 | 2.92E-14 |
|  | 30bus | 1.31E-13 | 1.17E-10 | 8.41E-14 | 0.00E+00 | 1.11E-11 |
|  | 57bus | 6.76E-11 | 8.51E-08 | 2.58E-12 | 1.48E-05 | 1.09E-05 |
| DNN | 6bus | 2.72E-06 | 7.04E-06 | 2.93E-05 | 9.70E-08 | 1.01E-07 |
|  | 30bus | 4.08E-07 | 1.86E-06 | 5.84E-05 | 0.00E+00 | 6.00E-09 |
|  | 57bus | 2.44E-04 | 5.90E-05 | 5.21E-03 | 1.19E-08 | 2.58E-06 |
| Gurobi | 57bus | 4.01E-13 | 8.63E-10 | 5.96E-15 | 0.00E+00 | 5.13E-19 |

Table 2: Mean Squared KKT Errors on test set generated for a realistic demand range

We calculated squared distance between the true optimal conditions and the model predictions as KKT 1, 2 and 4 are 0, and KKT 3 is non-positive at optimality.

Table 2 reports average KKT violations for the three models on the test sets with realistic load characteristics. Note that this dataset has the same distribution characteristics as the training set for classical DNN training but is out-of-distribution for SSNN. We report only the 57-bus system Gurobi results due to page limitations, but as expected, the Gurobi model outperformed both NN based approaches on most KKT measures. The SSNN predictions satisfied all KKT conditions with less than 1e-10 squared error for 6- and 30-bus systems, with relatively stable performance across the three test systems. The only exception is that KKT 3 and 4 calculations are reduced to around 1e-5 in 57 bus system. On the other hand, KKT performance of the classically trained DNN models for 6- and 30-bus are close but is lower for 57-bus. KKT 1 MSE increased from 2.72e-06 and 7.04e-06 for 6-bus to larger than 1e-5 for 57-bus. Similar performance reductions can be seen for the KKT 2 and KKT 4 results, while the KKT 3 results were relatively stable with respect to the system size.

Table 3 reports average KKT violations on test sets with extreme load characteristics, reflecting out-of-training distribution examples for classical DNN and mostly out-of-training distribution for SSNN[2]. Performance on the extreme datasets was close to the realistic demand results for SSNN, with 6- and 30-bus errors less than 1e-10 for all KKT measures. Performance reduced somewhat for the 57-bus system, where KKT 1 ($\boldsymbol{\delta}$), KKT 3 and KKT 4 errors increased to 4.07e-08, 2.76e-8 and 2.62e-8 respectively. For this dataset, KKT violations of DNN performance is well above SSNN. Best performance was recorded for 30-bus system which produced at least $10^2$ times more squarred error.

|  |  | KKT 1 ($\mathbf{P}^g$) | KKT 1 ($\boldsymbol{\delta}$) | KKT 2 | KKT 3 | KKT 4 |
|---|---|---|---|---|---|---|
| SSNN | 6bus | 6.03E-13 | 1.61E-10 | 1.52E-13 | 1.40E-14 | 2.64E-14 |
|  | 30bus | 1.64E-13 | 1.73E-10 | 2.04E-13 | 2.69E-12 | 1.20E-10 |
|  | 57bus | 1.98E-11 | 4.07E-08 | 3.07E-12 | 2.76E-08 | 2.62E-08 |
| DNN | 6bus | 7.15E-03 | 8.38E-01 | 3.26E+00 | 1.66E-02 | 1.48E-03 |
|  | 30bus | 1.66E-02 | 1.54E-02 | 3.82E-02 | 4.76E-05 | 2.32E-07 |
|  | 57bus | 7.41E+01 | 8.50E+01 | 3.28E+01 | 9.75E-04 | 1.24E+00 |
| Gurobi | 57bus | 7.88E-13 | 1.36E-09 | 1.76E-14 | 0.00E+00 | 1.30E-15 |

Table 3: Mean Squared KKT Errors on test sets generated for the extreme demand characteristics

---

[2]Recall that this dataset was sampled by varying one bus load while fixing others to 0.01 and repeating for all buses. Therefore, it is mostly out-of-training distribution for SSNN, which was trained with a dataset that allows for variations in two load buses only

## 4.2 Uncertainty

A comparison of the prediction speed of our SSNN model against Gurobi showed that our model has significant computational advantages over traditional solvers. To calculate solutions for 5000 observations, SSNN was 63 times faster than the Gurobi solver for the 6 bus system, which increased to 3543 times faster for the 57 bus system. In nominal terms, the model takes 0.01 to 0.05 seconds to calculate all solutions for all test systems, while solution times for Gurobi increased substantially with system size.

The low computational cost and strong KKT performance of the SSNN model implies that it can be used to quickly generate large distributions of optimal and feasible solutions for the simulation and long term planning of energy systems with uncertain renewable resources, such as wind generation. To explore this we modified eq. 9b of the DC-OPF problem as follows to allow variation:

$$\mathbf{B}\boldsymbol{\delta} - \mathbf{P}_g - \mathbf{P}_{ren} + \mathbf{P}_d = 0, \tag{10}$$

where $\mathbf{P}_{ren} \sim \text{Exponential}(\lambda = 1.25)$. In the simulations, the problem was solved repeatedly for 500 random cases of $\mathbf{P}_{ren}$ for each hour. A 1.5 unit generator limit was enforced by truncating the generated values above 1.5.

For this analysis, default load values from the Pandapower package were taken as base demand. To reflect seasonality of hourly changing demand, we used historical demand data for Ontario, Canada [3] as a scaler. Specifically, an arbitrary day (May 11, 2024) was selected and the daily demand was divided by the maximum value throughout the day to act as scalers. Each bus value was multiplied by the constant for every hour to obtain 24 demand inputs. To calculate solutions for 1200 different inputs, our SSNN model ran for 0.03 seconds.

Figure 6 in Appendix A.4 presents the distribution of the resulting 500 optimal generator dispatch for 24 hours using boxplots. All three generator have a capacity limit of 0.8 mW. As shown, the boxplots for the lowest cost Generator 2 are higher than other generators. During the day, Generator 2 often runs at full capacity and the dual variable of the associated capacity constraint ($P_2 - 0.8 \leq 0$) is positive. During peak hours (5pm-9pm), Generator 1 also often reaches capacity with its dual variable increasing above zero, forcing the system to use Generator 3.

## 5 Conclusion and Future Research

This paper proposes an explainable semi-supervised NN modeling approach for multiparametric QP optimization problems with linear constraints that provides precise solutions with generalizability power. The model aligns the piecewise linear nature of NN architecture with the underlying mathematical structure of the optimization problem by deriving the model weights directly from the linear sensitivity function of solutions in each critical region defined by the sets of binding constraints. To train the model, a cost efficient data generation approach was proposed that bypasses using solvers in a loop. As a proof of concept, the SSNN modeling approach was applied to DC-OPF problems with upper and lower bound generator limits. SSNN models trained on only 1000 data points achieved higher KKT optimality and feasibility results than generic DNN models trained classically on five times as much data. The SSNN models generated optimal solutions to large input sets much faster than the Gurobi solver, with little sacrifice in KKT performance. As such, the models can rapidly create a distribution of optimal and feasible solutions to inputs sampled from empirical distributions of uncertain demand and renewable generator dispatches.

This work has focused on the SSNN model architecture, its training, and providing proof-of-concept support. We acknowledge that further work is needed to develop an efficient discovery algorithm for identifying active constraint combinations that is applicable to a general class of empirical applications. Our results indicated some declining performance with the larger DC-OPF systems. Further work is also needed to identify the difficulties in training SSNN models to scale up to larger systems.

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

## A APPENDIX / SUPPLEMENTAL MATERIAL

### A.1 NNS AS PIECEWISE LINEAR FUNCTIONS

Our approach relies on the observation that shallow NNs with ReLU activation function fall under the class of PWL whose slope and intersection parameters are estimated from data. As a PWL function, NNs can estimate any continuous function by fitting $d_h + 1$ linear pieces. Thus, as $d_h$ increases, the estimation of the function improves given that the model is trained with a large dataset. However, more complex models, i.e. $d_h > n$, cannot guarantee to estimate a piecewise linear target function with $n$ connected line as it can zero the training error in infinitely many ways. To support the discussion in this paper, this section formulates NN as PWLs.

A shallow NN with $d_h$ neurons can be written as

$$f(\mathbf{x}, \theta) = \mathbf{W}^1 \sigma(\mathbf{W}^0 \mathbf{x} + \mathbf{b}^0)) + \mathbf{b}^1, \tag{11}$$

where $\theta = [\mathbf{W}^0, \mathbf{W}^1, \mathbf{b}^0, \mathbf{b}^1]$ is the parameter set, $\mathbf{W}^0 \in \mathbb{R}^{d_h \times d_x}$, $\mathbf{W}^1 \in \mathbb{R}^{d_y \times d_h}$, $\mathbf{b}^0 \in \mathbb{R}^{d_h}$, $\mathbf{b}^1 \in \mathbb{R}^{d_y}$, $\sigma(\mathbf{z}) = ReLU(\mathbf{z})$, $\mathbf{x} \in \mathbb{R}^{d_x \times d_d}$, with $d_x, d_y$ are input and output dimensions of the model, respectively, and $d_d$ is the number of observations.

For the case of a univariate function, $d_x = d_y = 1$, with $d_h$ number of neurons in the hidden layer, the model can be written as

$$f(x_i, \theta) = \sum_{j=1}^{d_h} W_j^1 (W_j^0 x_i + b_j^0)^+ + b^1, \tag{12}$$

where $x_i$ is the $i$th observation in the dataset, $(z)^+ = z$ if $z > 0$ and 0 otherwise.

It can be seen that the above function is a weighted summation of hidden layer output, which generates a different slope for any interval of $x_i$ depending on whether $(W_j^0 x_i + b_j^0) > 0$ in that interval. To express mathematically, let $\mathcal{B}_i = \{j \in [1, \ldots, d_h] \mid (W_j^0 x_i + b_j^0) > 0\}$ be the set of indices of active slopes in the interval, $x^{i-1} \le x \le x^i$. Then we can write,

$$f(x_i, \theta) = b^1 + \begin{cases} 0 & \text{for } x_i \in (-\infty, x^1) \\ \sum_{j \in \mathcal{B}_1} W_j^1 W_j^0 x_i + W_j^1 b_j^0 & \text{for } x_i \in [x^1, x^2) \\ \vdots & \vdots \\ \sum_{j \in \mathcal{B}_{n-1}} W_j^1 W_j^0 x_i + W_j^1 b_j^0 & \text{for } x_i \in [x^{n-1}, x^n) \\ \sum_{j \in \mathcal{B}_n} W_j^1 W_j^0 x_i + W_j^1 b_j^0 & \text{for } x_i \in [x^n, \infty) \end{cases} . \tag{13}$$

Notice that, for each input interval, NN can have a different slope. Also, the maximum number of slopes that a model can have is limited by $d_h + 1$ due to the nature of ReLU function that is activated within certain range but is not deactivated afterwards with an increase of the input variable.

By substituting $M_i = \sum_{j \in \mathcal{B}_i} W_j^1 W_j^0 x_i$, $N_i = \sum_{j \in \mathcal{B}_i} W_j^1 b_j^0$, one can obtain the general form of a PWL function,

$$f(x_i, \theta) = \begin{cases} N_0 & \text{for } x_i \in (-\infty, x^1) \\ M_1 x_i + N_1 & \text{for } x_i \in [x^1, x^2) \\ \vdots & \vdots \\ M_{n-1} x_i + N_{n-1} & \text{for } x_i \in [x^{n-1}, x^n) \\ M_n x_i + N_n & \text{for } x_i \in [x^n, \infty) \end{cases} . \tag{14}$$

Figure 5 illustrates the NN as a piecewise linear function. As illustrated, for $x_3 < 2$, none of the hidden neurons have a positive value as the $\mathbf{W}_j^0 x + b_j < 0$ for all $j$. In the interval $2 \le x_3 < 4$, the set of active neurons is $\mathcal{B}_1 = \{1\}$ as only $\mathbf{W}_1^0 x + b_1 > 0$ and the output increases with a certain slope as $x_3$ increases. After $x_3 \ge 4$, another slope is also activated and the set of active slopes are $\mathcal{B}_2 = \{1, 5\}$, which updates the change of the output value with respect to $x_3$.

If the target function is not a PWL function but an arbitrary nonlinear continuous function defined in a bounded interval, $x_i \in [a, b]$, the NN can be used to approximate the function, and the accuracy of the predictions increases as $d_h \to \infty$, as stated in to Universal Approximation Theorem (UAT) Hornik et al. (1989). Note that the UAT confirms the theoretical existence of a NN model representing a continuous target function within an error margin, but does not specify how such a network can be trained. In practice, model performance depends also on size of the training set and whether the boundary points are included in the set. This makes it infeasible to achieve a desired error level for target functions with larger input/output size as as number of boundary points, $2^n$, increase dramatically as $n$ increase.

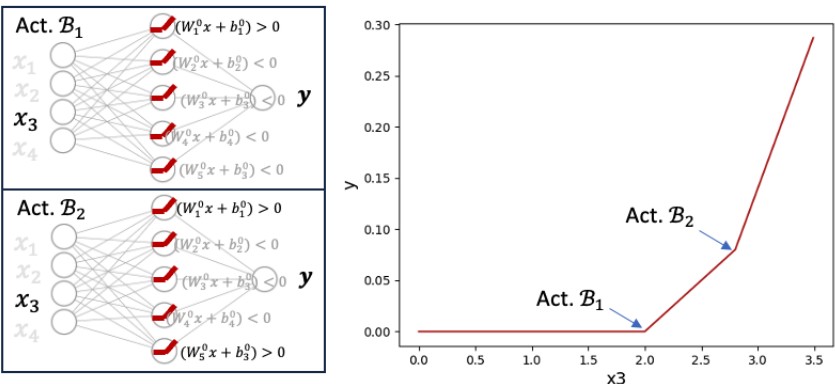

Figure 5: NN as a Piecewise Linear Function

## A.2 DERIVATION OF EQUALITY CONSTRAINED PROBLEM SOLVER

The gradient of the Lagrangian function (3) with respect to $\mathbf{x}$, $\boldsymbol{\lambda}$ must be zero at optimality as follows:

$$\frac{\partial L}{\partial \mathbf{x}} = 2\mathbf{Q}\mathbf{x} + \mathbf{C} - \boldsymbol{\lambda}^T \mathbf{A}_e = 0 \tag{15a}$$

$$\frac{\partial L}{\partial \boldsymbol{\lambda}} = \mathbf{b}_e + \mathbf{F}_e \boldsymbol{\theta} - \mathbf{A}_e \mathbf{x} = 0. \tag{15b}$$

Since, the parameter $\boldsymbol{\theta}$ in (15b) can change due to the variability of the studied problem, such as electricity demand in power system application and non-dispatchable supplies such as wind and solar generators, our goal is to train a model that predicts the optimal solution of (1a)-(1b) corresponding to an arbitrary $\boldsymbol{\theta}$, while the remaining system parameters $\mathbf{Q}, \mathbf{C}, \mathbf{A}_e, \mathbf{F}_c, \mathbf{A}_e, \mathbf{A}_{\mathcal{C}}$ are considered constant. Therefore, equations (15a) and (15b) can be written in matrix form as,

$$\begin{bmatrix} 2\mathbf{Q} & -\mathbf{A}_e^T \\ -\mathbf{A}_e & 0 \end{bmatrix} \begin{bmatrix} \mathbf{x} \\ \boldsymbol{\lambda} \end{bmatrix} = \begin{bmatrix} \mathbf{e}_1 - \mathbf{C} - \mathbf{F}_c \boldsymbol{\theta} \\ \mathbf{e}_2 - \mathbf{b}_e - \mathbf{F}_e \boldsymbol{\theta} \end{bmatrix}, \tag{16}$$

where $\mathbf{e}_1, \mathbf{e}_2$ are mismatch variables, where in a linear system, the optimal solution to (1a)-(1b) for a given $\boldsymbol{\theta}$ can be found by setting $\mathbf{e}_1, \mathbf{e}_2 = 0$, which can be obtained by solving (16). To this end, the $\mathbf{J}$ matrix can be defined as below, whose inverse is used to find the optimal solution as

$$\mathbf{J}^{-1} \begin{bmatrix} -\mathbf{C} - \mathbf{F}_c \boldsymbol{\theta} \\ -\mathbf{b}_e - \mathbf{F}_e \boldsymbol{\theta} \end{bmatrix} = \begin{bmatrix} \mathbf{x}^* \\ \boldsymbol{\lambda}^* \end{bmatrix}, \quad \text{for} \quad \mathbf{J} = \begin{bmatrix} 2\mathbf{Q} & -\mathbf{A}_e^T \\ -\mathbf{A}_e & 0 \end{bmatrix}. \tag{17}$$

From the above equation 17, the function $g(\mathbf{C}, \mathbf{b}_1)$ can be obtained as

$$g(\mathbf{C}, \mathbf{b}_1) = \begin{bmatrix} 2\mathbf{Q} & \mathbf{A}_1^T \\ \mathbf{A}_1 & 0 \end{bmatrix}^{-1} \begin{bmatrix} -\mathbf{C} \\ -\mathbf{b}_1 \end{bmatrix}. \tag{18}$$

## A.3 DERIVATION OF LAGRANGIAN DERIVATIVES FOR DC-OPF

$$L(\mathbf{P}_g, \boldsymbol{\theta}, \boldsymbol{\lambda}, \boldsymbol{\mu}) = \mathbf{P}_g^T \mathbf{Q} \mathbf{P}_g + (\mathbf{C}^T + \mathbf{F}_c \boldsymbol{\theta}) \mathbf{P}_g + \mathbf{C}_0 +$$
$$\boldsymbol{\lambda}(\mathbf{P}_d - \mathbf{P}_g - \mathbf{B}\boldsymbol{\delta}) + \boldsymbol{\mu}^+ (\mathbf{P}_g - \mathbf{P}_g^+) + \boldsymbol{\mu}^- (\mathbf{P}_g^- - \mathbf{P}_g). \tag{19}$$

Following the same steps in Section 3, our model is derived using the derivatives of the Lagrangian function of the DC-OPF problem can be derived as below

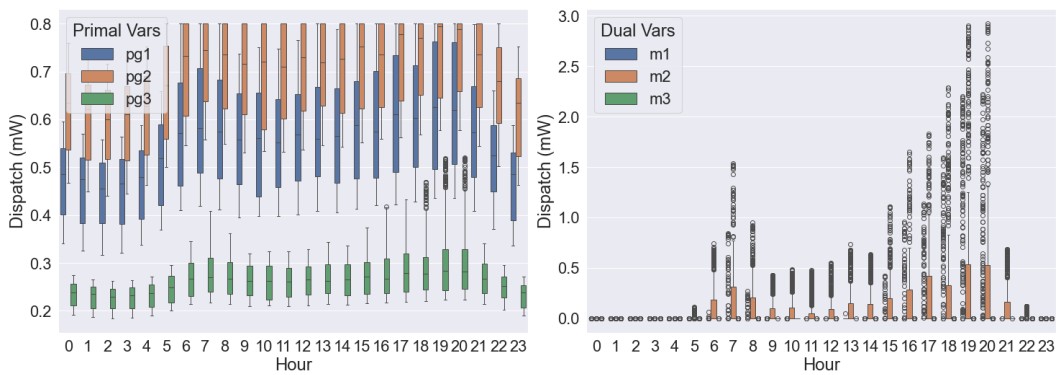

Figure 6: Hourly distribution of optimal primal and dual prices

The partial derivatives of the above function is

$$\frac{\partial L}{\partial \mathbf{P}_g} = 2\mathbf{Q}\mathbf{P}_g + \mathbf{C}_1 + \boldsymbol{\lambda} + \boldsymbol{\mu} = 0, \tag{20a}$$

$$\frac{\partial L}{\partial \boldsymbol{\theta}} = \mathbf{B}^T \lambda = 0, \tag{20b}$$

$$\frac{\partial L}{\partial \boldsymbol{\lambda}} = \mathbf{P}_d - \mathbf{P}_g - \mathbf{B}\boldsymbol{\delta} = 0, \tag{20c}$$

$$\frac{\partial L}{\partial \boldsymbol{\mu}} = \mathbf{P}_g - \mathbf{P}_g^{max} = 0. \tag{20d}$$

### A.4 Hourly distribution of optimal primal and dual prices

See Figure 6.

### A.5 Further Details on Training of SSNN

All models were compared on two types of datasets, realistic and extreme. The former was constructed by multiplying a base load with a factor drawn from Uniform(0.6,1.4). The latter was constructed by choosing one $\theta_i$ and sampling $m$ values from Uniform(0,*max*) while fixing the remaining $\theta_j$ to 0.01 and repeating this for all parameters. For this method, *max* was set to the sum of all generator capacity limits. To generate $M$ number of data points, $m$ is set to $\lceil M/n_l \rceil$ where $n_l$ is the number of dimensions. If more than $M$ points were sampled, the excessive amount was removed.

While SSNN was tested on these datasets, the training set is constructed using the critical regions $[\mathbf{CR}_i^-, \mathbf{CR}_i^+]$ detected via the discovery step. To discover sets of binding constraints, all parameters were set to 0.01 and an arbitrary $\theta_i'$ was incrementally increased from 0 to the maximum number the system allows. The discovery was carried out on one dimension only, so this approach would not detect if other sets of binding constraints could be found by repeating this step with other $\theta_i$.

To construct the training set for SSNN, different values of $\theta_i'$ were sampled from each $[\mathbf{CR}_i^-, \mathbf{CR}_i^+]$ and solved by expanding the coefficient matrix, $\mathbf{J}_{\mathcal{B}_)}$ with the corresponding binding constraints, running the calculations in eq. 8 and checking if the solutions satisfy the KKT conditions. This step was repeated using one other $\theta_j$ and fixing all other parameters, assuming the same $[\mathbf{CR}_i^-, \mathbf{CR}_i^+]$ would apply to this range. If the calculated solution does not satisfy KKT, another number was sampled from the same interval.

The problem of critical regions that do not generalize to other dimensions could be solved by calculating optimal solutions using all discovered $\nabla \boldsymbol{\mu}^*$ and choosing the ones that satisfies KKT conditions. For this study, however, our models were trained using the above mentioned approach.

