# OpenReview forum: "Semi-Supervised Neural Network Model For Quadratic Multiparametric Programming"
_ICLR.cc/2025/Conference — Submitted to ICLR 2025_

### Official Review · Reviewer_vpgz · 2024-11-02

**Soundness:** 2
**Presentation:** 2
**Contribution:** 2
**Rating:** 3
**Confidence:** 5

**Summary:**

The paper proposes a semi-supervised neural network (SSNN) architecture designed to solve multiparametric quadratic programming (mp-QP) problems with linear constraints. The authors develop an SSNN that tries to incorporates the mathematical structure of the global solution function from the Langrangian gradient. It is claimed that proposed SSNN derives a significant portion of its weights analytically from the problem's characteristics and hence data requirement is less with higher accuracy.

**Strengths:**

1. The studied problem is interesting and be of significant in the community.
2. The idea is straightfoward and reasonable to leverage the known gradient information of the problem.

**Weaknesses:**

I have several concerns about the main claims in the paper:

1. The theoretical result is questionable. Please see below
2. The reason that fix the first layer weights is unclear, leading the DNN design is questionable. Please see below
3. The literature review is quite insufficient and many related work are missing, leading the claim of the paper contribution unsolid. Please see below.
4. The presentation can be further improved. Please see below
5. The comparison is insufficient, the simulation results lack solid baseline support. Please see below

**Questions:**

1. It is well known that even with primal problem is strictly convex with unique solution, the dual variables may not be unique for general optimization. How do the authors derive that \mu(\theta) is piecewise linear as \mu themself may not be unique? Please provided the detailed proof.

2. Even the gradient is known, why the first layer weights should be fixed? The ReLu activations and all other remaining DNN parameters can easily lead the final DNN function with a different gradient. In addition, fixing the weights will also decrease the number of free parameters of DNN, and break the universal approximation of the DNN. Please clarify.

3. Many related papers are missing, the authors are suggested to do a more comprehensive literature survey, e.g.,
[R1] Zhang, Ling, Yize Chen, and Baosen Zhang. "A convex neural network solver for DCOPF with generalization guarantees." IEEE Transactions on Control of Network Systems 9.2 (2021): 719-730.
[R2] Donti, Priya L., David Rolnick, and J. Zico Kolter. "DC3: A learning method for optimization with hard constraints." arXiv preprint arXiv:2104.12225 (2021).
[R3] Pan, Xiang, et al. "Deepopf: A deep neural network approach for security-constrained dc optimal power flow." IEEE Transactions on Power Systems 36.3 (2020): 1725-1735.
[R4] Li, Meiyi, Soheil Kolouri, and Javad Mohammadi. "Learning to solve optimization problems with hard linear constraints." IEEE Access 11 (2023): 59995-60004.

4. There have been several parameters undefined as presented, e.g., b1 at line 215, \mu-DNN at 255.

5. The simulation results are insufficient and the details are not presented. The simulation lacks the final solution feasibility (percentage) result and the optimality performance. In addition, it only compares with the most naive DNN, which can not make the advantage of the approach clear.

---

### Official Review · Reviewer_DWwj · 2024-11-03

**Soundness:** 2
**Presentation:** 2
**Contribution:** 2
**Rating:** 3
**Confidence:** 4

**Summary:**

This paper targets the DC Optimal Power Flow (DC-OPF) problem for power transmission systems. A Semi-Supervised Neural Network model is proposed to predict the optimal solutions of DC-OPF. The model's efficacy is demonstrated on several IEEE standard transmission systems, compared to regular DNN and conventional optimization solver (Gurobi).

**Strengths:**

•	By embedding the physical properties of power systems (such as system topology and DC-OPF formula) directly into the neural network’s first layer, the approach enhances interpretability and offers a data-efficient solution.

•	The determination of critical region detection with binding constraints is novel and makes the SSNN-based solution prediction efficient.

•	The use of standard IEEE transmission systems (IEEE-6, IEEE-30, IEEE-57) validates the model’s applicability in real-world power system problems.

**Weaknesses:**

•	The proposed method seems specifically applicable to DC-OPF, where the constraints are linear. However, DC-OPF is a linearization of the AC-OPF problem, where the power flow constraints are nonlinear and nonconvex. Since SSNN is a data-driven solver, can it be generalized to broader problems? Otherwise, how to deal with the approximation errors?

•	One claimed contribution is reducing training data, but the data acquisition procedure is unclear. Based on Sec. 3.4 and 4.1, are the data prepared by extensively solving the DCOPF under different settings using conventional solvers? If so, the motivation can be problematic. For data efficiency, why not use ML models with less complexity? Thus, the baseline is very limited, where only regular NN is used for comparison. Moreover, the initial use of a solver to set weights introduces a dependency, even though it can reduce the demand for training data relatively. Clarifying the trade-offs here would be beneficial.

•	The extrapolation challenge identified in this work is essential for applications in power systems. However, despite SSNN’s performance on out-of-distribution tests being strong, the paper lacks a theoretical explanation for this robustness.

**Questions:**

Please refer to the bullet points in Weaknesses.

---

### Official Review · Reviewer_3ZeM · 2024-11-03

**Soundness:** 2
**Presentation:** 2
**Contribution:** 2
**Rating:** 3
**Confidence:** 3

**Summary:**

This paper proposes a semi-supervised neural network (SSNN) architecture for solving multiparametric quadratic programming (mp-QP) problems. The proposed approach derives certain model weights analytically from the mathematical structure of the global solution function. In tests on DC optimal power flow (DC-OPF) problems, the proposed method achieves superior performance, as measured by KKT conditions, compared to a deep neural network baseline. At inference time, the proposed method is shown to generate optimal solutions orders of magnitude faster than commercial solvers.

**Strengths:**

1. While this paper is not the first to point out the connection between mp-QP problems and the piecewise linear nature of neural networks with ReLU activation functions, the proposed method appears original and well-motivated mathematically.
2. The experimental results, although somewhat limited in scope, are very good. In particular, the proposed method remains accurate when tested on highly out-of-distribution data, whereas the standard DNN baseline becomes unreliable.
3. The use of IEEE test cases provides a good, fairly nontrivial validation of the method, although I have questions about its applicability to real-world conditions (see below).

**Weaknesses:**

### Major
1. In Section 2, the authors cite a reasonable number of works that are related to this paper either in terms of method (using NNs for optimization) or application (DC-OPF). However, I find the discussion of each individual method insufficient. The related work section would be vastly improved by a significantly more detailed discussion of a handful of key papers, emphasizing in particular how they relate to and are improved by the current work.

2. Similarly, the authors do not experimentally compare their method to any of the related work, except for a basic DNN baseline. It is hard to adequately evaluate the proposed method without a proper empirical comparison with the related work.

3. The method provides no theoretical guarantees about the optimality and feasibility of the predicted solutions. One way of compensating for this might be to include extensive empirical evidence for the effectiveness of the method in diverse settings. However, the experiments consist of a case study from a single domain which, although it is repeated at different scales, is not sufficient evidence to draw general conclusions about the soundness of the proposed method for mp-QP. I think the paper would be more convincing with at least one challenging experiment other than DC-OPF.

4. A significant limitation of this method, acknowledged by the authors, is the need to filter active constraints. The authors claim that, “in many practical mp-QP applications, characteristics of the problem imply that most constraint combinations can never occur,” but this claim is not supported, except for the given case study of DC-OPF. The generalizability of the constraint filtering approach to other mp-QP applications remains uncertain and should be addressed.

5. Relatedly, accurate estimation of the boundaries of critical regions appears to be critical to the correctness of the method. Please include a more detailed discussion of this aspect of the method and, in particular, its reliability.

6. The DC-OPF case study (Eq. 9) is poorly motivated and explained. Firstly, the variables $\bf{P_d}$, $\bf{F_e}$, and $\bf{B}$ are not defined, although presumably $\bf{B\delta}$ are losses. Secondly, we are not told what the parameters $\bf{\theta}$ are actually supposed to represent in this case study, even though the multiparametric nature of the problem is central to the proposed method. Thirdly, it’s not motivated why the costs of power generation would be quadratic; many of the related works use linear costs, for example. Please improve presentation and explain why the case study is relevant.

7. The paper would be significantly improved by a thorough analysis of the computational cost of the proposed method, broken down by each individual stage (including data generation, constraint filtering, training), and compared to classical solvers and other neural network approaches. As it stands, the authors mention a significant speedup at inference time, but overall it’s still not clear what advantage the proposed method offers over a classical solver or existing machine learning approaches, especially taking the entire computational cost into account. Please include a table detailing the computational cost of the method, including comparisons to related work.

### Minor
1. Given the complexity of the proposed method, the clarity of the paper would be improved by a pseudocode algorithm outlining the entire procedure.
2. A limitation of the method is that it only works for linear constraints, whereas some formulations of the OPF problem in the related work use nonlinear constraints.

**Questions:**

1. Of the related work, which methods are comparable to the proposed method, either in terms of mp-QP problems generally or DC-OPF specifically? If not comparable, why not?
2. Is it possible to provide firm theoretical guarantees about the optimality and feasibility of the predicted solutions?
3. What happens if the wrong critical region is identified, and hence the wrong active constraints are used? Is there any way of avoiding this possibility?
4. Why are the generation costs in the given formulation of the DC-OPF problem quadratic as opposed to linear or piecewise linear? What do the parameters $\bf{\theta}$ represent? More generally, how applicable is the case study to real power grid applications, given the various simplifications (e.g., DC vs AC, linear vs nonlinear power flow constraints)?
5. What are the total computational costs of the proposed method, broken down by each individual stage? How does this depend on the size of the problem and number of possible constraint combinations?
6. Have you tested the method on applications other than DC-OPF?

---

### Official Review · Reviewer_jSa6 · 2024-11-04

**Soundness:** 2
**Presentation:** 2
**Contribution:** 2
**Rating:** 3
**Confidence:** 4

**Summary:**

This study addresses challenges in applying neural networks (NNs) with ReLU activation functions to multiparametric quadratic programming (mp-QP) problems in engineering, especially energy management. While deep NNs have been used to model these problems by leveraging their piecewise affine properties, traditional approaches often fail to deliver accurate or feasible solutions, even with large datasets. To improve performance, the researchers propose a semi-supervised neural network (SSNN) architecture that integrates the mathematical structure of the global solution function into the model. Unlike conventional training, the SSNN determines the first layer of its model weights from the system's physical properties (sensitivities of dual variables). The proposed method is validated on a DC-Optimal Power Flow (OPF) problem.

**Strengths:**

This is an interesting paper presenting a method for learning solutions to multiparametric quadratic optimization problems.
The method is validated on DC-Optimal Power Flow (OPF), which represents an important problem in the power systems domain.

**Weaknesses:**

Related work:
- authors are encouraged to expand the related work section. Specifically, it would be interesting for the reader to highlight similarities and differences from other feasibility restoration layers or differentiable optimization layers that can ensure the satisfaction of constraints in learning to optimize settings. Also, there are several recent publications on solving OPF problems via learning solutions to the underlying multi-parametric programming problems, including nonlinear formulations.
Formal aspects:
- I am not sure that Theorem 1 and Lemma 1 provide any added value for the reader. They are completely disconnected from the text and are not used to argue for, certify, or motivate the proposed method.
Technical aspects:
- The proposed method could be presented in a clearer and more detailed manner.  The flow of equations is interrupted by text that does not provide sufficient clarity on the proposed algorithm. The notation is rather confusing, with variables and many subscripts that are not properly defined or thoroughly explained. For instance, Figure 2 indicates that the first layer weights W_0 are initialized as gradients of the  dual variables \mu, however, in the following text of section 3.4 they are given as W_0 = [\mu^*]
- The core of the method is the introduction of pre-calculated and fixed first-layer weights of the Relu deep neural network. These weights are calculated based on sensitivities of the dual variables \mu w.r.t. the parameters \theta. From the given description, it is hard to easily understand exactly how these weights are calculated. I believe the clarity of the paper can be substantially improved by explicitly formulating the steps of the algorithm in a more formal manner.
- The difference between Figures 2 and 3 and the associated text is not very clear. Are figures 3 and 4 just special cases of the method associated with the DC-OPF problem (9)?
- The method is validated only on a single problem, DC-OPF. With problem-specific modifications, the paper fails to demonstrate the generality as indicated in the methods section. Moreover, the selected instances of the selected DC-OPF problem are small academic examples and do not represent challenging large-scale optimization problems.

**Questions:**

Learning to predict active constraints and associated dual variables of optimization problems via supervised learning is known to be challenging due to an unbalanced dataset, i.e., there are many more instances without active constraints than those with active constraints. Did the authors encounter similar issues in their case study?

---

### Meta-Review · Area_Chair_ZHuR · 2024-12-17

**Metareview:**

This paper presents a semi-supervised neural network (SSNN) architecture for solving multiparametric quadratic programming (mp-QP) problems, demonstrated on DC Optimal Power Flow applications. By analytically deriving a portion of the network’s weights from the underlying mathematical structure, i.e., the system’s global solution function and dual variable sensitivities, the SSNN reduces data requirements while improving accuracy and feasibility. The authors demonstrate that compared to conventional deep neural networks and commercial solvers, the proposed SSNN achieves better KKT performance metrics and generates optimal solutions significantly faster at inference time.

**Strengths:** The reviewers found the paper's problem of interest, DC-OPF, to be potentially significant in the power systems community and the method to be well-grounded.

**Weaknesses** The reviewers raised several concerns about the work, including a lack of clarity, insufficient comparison with related research in the field, and a narrowly focused experimental scope.

The reviewers unanimously agreed that the paper in its current form is not ready for publication at ICLR due to the raised weaknesses. The reviewers also raised a number of questions that the authors did not reply to. Hence, I concur with the reviewers and vote for rejecting the paper.

**Additional Comments On Reviewer Discussion:**

The authors did not provide a response to the reviewers. Given the unanimous recommendation for rejection, no post-rebuttal discussion was necessary.

---

### Decision · Program_Chairs · 2025-01-22

Reject